# Key Factors for “Fishing” NTCP as a Functional Receptor for HBV and HDV

**DOI:** 10.3390/v15020512

**Published:** 2023-02-12

**Authors:** Huan Yan, Chunli Wang

**Affiliations:** State Key Laboratory of Virology, Institute for Vaccine Research and Modern Virology Research Center, College of Life Sciences, TaiKang Center for Life and Medical Sciences, Wuhan University, Wuhan 430072, China

**Keywords:** HBV, HDV, preS1, NTCP, receptor, viral entry, cross-linking

## Abstract

About ten years ago, Wenhui Li’s research group in China identified the sodium taurocholate co-transporting polypeptide (NTCP), a bile acid transporter predominantly expressed in the liver, as a functional receptor for hepatitis B virus (HBV) and its satellite hepatitis delta virus (HDV) through biochemical and genetic studies. This finding unraveled a longtime mystery in the HBV field and led to the establishment of efficient and easy-to-use HBV infection models, which paved the way for the in-depth study of the HBV entry mechanism and facilitated the development of therapeutics against HBV and HDV. The whole picture of the complex HBV entry process became clear upon the follow-up studies over the years, including the recent resolution found for the NTCP structure. As one of the first authors of the 2012 eLife paper on NTCP identification, here, I (H. Y.) share our experience on the bumpy and exciting journey of receptor hunting, particularly on the photo-cross-linking study and some detailed descriptions of the “fishing” process and summarize the key factors for our successful receptor identification. This review may also provide helpful insights for identifying a protein target by peptide or protein baits through cross-linking and immunoprecipitation.

## 1. Introduction

HBV chronic infection remains a tremendous public health burden to humans, especially in developing countries. More than 296 million people worldwide are infected with HBV and suffer from HBV-related liver diseases, such as acute and chronic hepatitis, cirrhosis, liver failure, and hepatocellular carcinoma, resulting in an estimated 820,000 death annually worldwide (WHO, 2022). HBV vaccinations are effective, but we remain unable to eradicate the disease from most infected individuals. Around 15 million HBV patients are also co-infected with the hepatitis D virus (HDV) and suffer from a greater risk of rapid progression and severe disease, including cancer and liver failure [1].

HBV is a small DNA virus with a circular genome of about 3.2 kilobase pairs (kb) that was discovered in 1965 by Dr. Baruch Blumberg [2]. HDV is a satellite virus that uses HBV envelope proteins for viral assembly and cellular entry and, thus, can be employed as an ideal surrogate for studying HBV entry. HBV encodes small (S), middle (M), and large (L) multiple-transmembrane envelope proteins for the formation of outer coats. These proteins share the same S domain sequences as the carboxyl-terminal (C-terminal) region but with different lengths of amino-terminal (N-terminal) sequence extensions named preS1 and preS2, respectively [3]. Previous studies have demonstrated that the N-terminal preS1 domain of the L protein might play a central role in receptor recognition. Mutations in this region abolish viral infectivity, and the synthetic N-terminal myristoylated preS1 peptide encompassing the N-terminal region preS1 sequences specifically binds to the hepatocyte and efficiently blocks the HBV and HDV infection. Fine mapping of the preS1 region has highlighted the importance of the N-terminus lipidation and the N-terminal 2–48 amino acids (aa) for inhibitory activity, especially in the aa9–15 critical regions [4,5,6,7].

HBV and HDV are known to enter liver cells by binding to receptors on their surface before being engulfed. As human-adapted viruses, HBV and HDV exhibit a very narrow host tropism and only infect several types of freshly isolated primary hepatocytes from humans, such as chimpanzees (several other endangered primates) and treeshrew (*Tupaia belangeri*) [8,9,10]. How HBV specifically recognized these primary hepatocytes and gained access to these cells was largely unknown. Before 2012, HepaRG was the only reported cell line that was susceptible to HBV and HDV infections after culturing with a particular differentiation medium [11]. However, both primary hepatocytes and HepaRG are difficult to obtain and culture by most research groups. The lack of a feasible model for HBV infection significantly hampered the progress in HBV entry-related research, a situation that could be turned around by identifying the functional receptor of HBV. Many studies interested in the HBV receptor identified preS1 interacting proteins and considered them as putative HBV receptors. The incomplete list includes interleukin-6 (IL6) [12], human squamous cell carcinoma antigen 1(SCCA1) [13], the immunoglobulin A receptor (IgA R) [14], asialoglycoprotein receptor (ASGPR) [15], glyceraldehyde 3 phosphate dehydrogenase (GAPDH) [16], nascent polypeptide-associated complex a polypeptide (NACA) [17], glucose-regulated proteins (GRP75) [18], lipoprotein lipase [19], and four PreS1 binding proteins with an unknown identity and apparent molecular weight of 30 kDa [20], 47 kDa (HBV-BP) [21], 50 kDa (HBV-BF) [22], and 80 kDa (P80) [23], respectively. However, none of the proposed putative receptors can confer susceptibility to non-permissive cells through exogenous expression, and none of these studies further investigated the post-binding events mediated by these proteins.

In 2012, Yan et al. identified NTCP as a functional receptor for HBV and HDV based on studies of a preS1 peptide-mediated near-zero distance photo-cross-linking and tandem affinity purification [24]. The work of identifying NTCP as the functional HBV receptor was led by Wenhui Li and was accomplished by a team spearheaded by Huan Yan with critical contributions from Guocai Zhong (co-first author) and many other colleagues at the National Institute of Biological Sciences, Beijing (NIBS, Beijing) in China. In this review, we summarize the important clues and critical factors for successful HBV receptor hunting and also discuss the possible reasons for the failure of previous attempts. To vividly illustrate the receptor hunting process, here we liken our teamwork to catching a special fish. The fishing pond, bait, hook, float, fishing line, fishing tricks, fish identification, and fish verification are elaborated to show details about how the “goldfish”—NTCP—was captured(Figure 1).

## 2. Fishing Pond

The first crucial step for catching a particular fish is to find out in which pond the goldfish lives. Similarly, an ideal “fishing pond” for receptor purification and identification should be cells with high receptor abundance, such as cells susceptible to HBV and HDV infection. However, it is worth mentioning that, mainly due to the lack of easily accessible HBV permissive cells, most previous efforts used non-permissive cells or serum components for receptor purification or verification, such as HepG2, Huh-7, lymphocytes, and human serum [12,15,16,18,21,23,25]. 

We wanted to use primary human hepatocytes, primary *Tupaia* hepatocytes (PTHs), or well-differentiated HepaRG cells as target cells for our receptor discovery endeavor. Considering the cost and availability, we used PTHs as the material for receptor identification and purification. Our team took between two and three years to establish and optimize the PTHs-based platform, which consists of a list of methods and techniques related to treeshrew feeding, liver perfusion, hepatocyte culture, and PTH-based viral infection assays. Based on this platform, we could readily prepare a relatively large amount of freshly isolated PTHs for subsequent HBV and HDV infection-related experiments. These fresh isolated PTHs had to be used in time since the high susceptibility status could only be sustained for about three days under the culture conditions we developed in the lab. Several team members also established a high-quality transcriptome database of the *Tupaia* hepatocytes, which provided critical support for our mass spectrometry-based analysis and genetic studies of the hepatocytes from this unconventional animal model.

We then performed a series of PTHs binding and infection-related experiments to characterize HBV and HDV entry. Of note, HDV shares the same envelope proteins as HBV but has several advantages in receptor-binding-related experiments. First, HDV reverse genetics for generating HDV mutants is easier than that of HBV in our hands. Second, HDV virion quantification is more accurate than that of HBV due to the presence of a large amount of noninfectious HBV nucleocapsids in the cell culture-derived HBV supernatant. Yan et al. obtained several important clues through HDV binding assays: HDV can bind more efficiently with PTHs than other tested non-susceptible cells; HDV carrying specific preS1 mutations around aa9–16 showed a reduced binding efficiency in PTHs; HDV binding could be inhibited with the pretreatment of the PTHs by the myristoylated preS1 peptides. These experimental data, together with other results, such as the preS1 peptide blocking assay of HBV infection on PTHs, confirmed that the PTHs could be used as an excellent fishing pond for HBV receptor hunting.

## 3. Bait and Hook

After targeting the fishing pond, preparing proper bait was crucial for successful fishing (Figure 2). Theoretically, the bait for receptor hunting can be designed based on HBV or HDV viral particles or viral components that specifically interact with the unknown receptor. We attempted various preS baits and finally chose the functional myristoylated HBV preS1 peptide as the prototype for bait design. Compared with viral particles or full-length L proteins, the smaller preS1 peptide was considered to be better exposed, easier for modification, and might have lower non-specific binding. Of note, previous studies have tried to use preS1 peptide or preS1 fusion proteins as bait but have failed to identify any functional HBV receptor through conventional immunoprecipitation (IP) methods [13,18,20,23]. Except for using the improper cells for receptor identification, the following possibilities may also have contributed to the failure of many previous attempts for receptor hunting. This includes the insufficient receptor-binding activity of the bait due to the sub-optimal bait design strategy; the expression of the receptor in the target cells could be insufficient; the binding between the peptide and the receptor might not be strong enough; and the virus–receptor interaction might be detergent sensitive. Accordingly, it is crucial to use a receptor binding activity-validated preS1 lipopeptide with a barbed hook to firmly hold the bait-biting fish. Live cell cross-linking can play this role by forming covalent bonds between the preS1 bait peptide and the receptor under physiological conditions, enabling Western blot (WB) detection and highly stringent and tandem purification procedures, regardless of the binding stability and structure integrity of the bait-target proteins complex. By contrast, conventional IP methods cannot achieve similar purification procedures, for which non-specific binding and artificial binding are of great concern. 

We chose photo-reactive unnatural amino acids (UAAs) rather than less specific commercial protein crosslinkers because the former could be placed at receptor-binding sites to achieve a higher cross-linking specificity. We initially attempted to choose DiZPK, a lysine backbone-based UAA with a diazirine ring that can produce a reactive carbene group upon 365 nm ultraviolet (UV) light irradiation. However, the structure of the binding interface was critical for specific interactions, so subtle alterations resulting from the DiZPK substitution could have severely interfered with the binding affinity. To minimize the potential interference, Huan Yan suggested using the L-photo-leucine (L-2-amino-4,4-azi-pentanoic acid), which is structurally similar to the canonic leucine with a very short spacer between diazirine and the main protein chain for higher specificity. Another reason for choosing photo-leucine was because there were four residues on the preS1 peptide that could be considered for substitution with photo-leucine: the sites 11 and 14 (HBV Genotype F and G contain Leu at site 14) within the core receptor binding region (aa9-15) and sites 5 and 19 close to the critical region. For cost-effectiveness, we picked sites 11 and 14 as substitutions for commercial peptide synthesis and left the four sites incorporation as a backup plan. We confirmed that the synthetic bait peptide with two photo-leucine substitutions well-preserved their PTH binding and infection-inhibitory activity. In contrast, a control bait peptide with similar sequences but with an N9K substitution completely lost the activity to inhibit the viral infection and HDV binding. In fact, this indispensable N9K control peptide arrived three months later than the wild-type (WT) peptide. Data based on this control peptide offered critical information showing us the right path for receptor hunting.

## 4. Float and Fishing Line

It is crucial to know whether the fish is biting the bait and when to pull the fish out. The photo-cross-linking approach labeled the target proteins with the bait peptide, enabling better detection and purification of the cross–linked complex through the tags on the bait peptide. A detection tag for WB can act as a “float” to monitor the cross-linking experiments, while an affinity tag for purification can be considered the “fishing line” to pull the fish out. For example, the efficiency and specificity of the cross-linking can be examined by WB detecting the tag on the bait as well as the bait–target complex. By contrast, the affinity tags on the bait peptide rendered the purification efficient.

In many cases, one tag can be sufficient for both detection and purification. However, we used two tags to maximize the utilization of the bait peptide, a C-terminus conjugated biotin and a native preS1 epitope (aa 21–33) beyond the critical receptor-binding region. To this end, a monoclonal antibody 2D3 recognizing this epitope was developed by the team. Indeed, the dual-tag strategy was superior to the single-tag strategy in many aspects of receptor identification. First, it enabled a more stringent sequential protein purification procedure named tandem affinity purification (TAP). Second, it facilitated the IP-WB protocol to effectively detect specific target signals from the background in biochemical analysis. In our case, using two chemically distinct tags (peptide epitope and biotin) could maximize the detection specificity and purification efficiency. Although both tags can be used for the two purposes, the 2D3 peptide epitope is better for detection, while the biotin moiety is more suitable for purification (Figure 2).

## 5. Fishing Tricks

As beginners, fishing, training, and practice were necessary for us to learn how to catch a big fish. There was no ready-to-use protocol for UAA-based photo-cross-linking assays, so we had to explore and test different conditions to establish a protocol. For example, the optimal peptide concentration, incubation temperature, UV irradiation distance and time, cell lysis and IP methods, and purification medium had to be determined beforehand. Before receiving the N9K control peptide, Huan Yan conducted many IP-WB experiments using the WT peptide alone and obtained several “promising” results, which later turned out to be artifacts. When incubating the WT peptide at the concentration of 10 μM, we observed a distinct cross-linking signal from PTH cell samples after UV irradiation (UV+ group) but not in samples without UV irradiation (UV-group), and the phenotype could only be observed in PTHs but not other tested cell lines. Unfortunately, we were very frustrated to find out that almost identical signal patterns could also be detected using the “N9K” control peptide under the same conditions three months later. Before giving up, Huan Yan wondered if he had put too much bait into the cells so that the non-specific cross-linking signal covered the specific signal. He then conducted a PTHs cross-linking experiment using serial-diluted bait peptides. Remarkably, the differences in the cross-linking signal, a smeared band around 65 kDa revealed by WB analysis, from WT and N9K peptides became evident when the concentration was reduced to 500 nM and more apparent at 250 nM. This important result showed the first specific NTCP cross-linking signal and uncovered that the optimal concentration is one of the critical conditions for obtaining the specific WB signal.

Yan et al. then tried many approaches to characterize the target protein, such as the features of glycosylation, oligomerization, phosphorylation, ubiquitination, and hydrophobicity. We found that the signal was mainly from a single N-glycosylated hydrophobic protein with an apparent molecular weight of ~43 kDa upon deglycosylation by PNGase F. This clue played a crucial role in subsequent receptor identification for several reasons. (1) Glycosylation is a common feature of cell surface receptors. (2) The phenotype of complete shift upon PNGaseF treatment indicates that the target signal was mainly from a single protein. In contrast, the phenotype was not observed in the signal obtained from N9K control peptides (under high peptide concentration). (3) The molecular weight with or without glycosylation could be used as a useful criterion to examine the protein candidates.

With successful experiences and “fishing tricks”, we proceeded to the goal of purifying enough target proteins for mass spectrometry (MS) analysis. We prepared a large number of PTHs and conducted photo-cross-linking under the established conditions. After testing different purification methods, Yan et al. applied a “biotin-antibody-biotin” tandem affinity purification strategy to enrich the bait–target complex proteins as much as possible. The 2D3-covalently conjugated magnetic beads (Dynabeads M-270 Epoxy) and Streptavidin-coupled magnetic beads (Dynabeads MyOne Streptavidin T1) for affinity purifications were selected for lower non-specific binding after several tests. As mentioned above, we took advantage of the glycosylation features for more rational sample preparation. According to the WB results, three silver staining gel bands were excised for MS analysis. Theoretically, the target protein should be present in the ~65 kDa gel band from the “WT-PNGase (-)” sample and the ~43 kDa gel band from the “WT-PNGase (+)” sample but not in the ~65 kDa gel band from the “N9K-PNGase (-)” sample. Now the fish we were looking for was probably in the bucket, although we still had no idea about the species name of the fish.

## 6. Fish Identification

To figure out the name of the fish we captured, we used the proteome database, which the lab established through the deep sequencing of the PTH transcriptome. The database could be considered as the fish catalog of the pond. By searching the MS data against this database, we obtained lists of protein hits from the three gel band samples, respectively. Although with a relatively low peptide coverage, NTCP (encoded by solute carrier family 10 members 1, *SLC10A1*) was one of the top candidates that drew our significant attention. Other proteins, such as albumin and protein disulfide-isomerase, were most likely not relevant to HBV infection.

We highly suspected that NTCP was the target protein we observed through WB for the following reasons. NTCP was presented as the fourth hit of sample “WT-PNGaseF(-)” and the second hit of sample “WT-PNGaseF(+)” but was not presented in the list of negative control sample “N9K-PNGaseF(+)”. NTCP is a glycosylated cell surface protein with a molecular weight that matches our target bands, either with or without glycosylation. One of the identified NTCP peptides (TEETIPGTLGNSTH) contains four *Tupaia* NTCP-specific residues (NSTH), indicating that the peptide is not from the contamination of human samples. NTCP is predominantly expressed in hepatocytes and not expressed in most hepatoma cell lines, which is consistent with the cell tropism of HBV. NTCP is a highly hydrophobic protein with nine predicted transmembrane motifs, which is difficult to purify by conventional IP methods. Collectively, all experimental evidence and unique features of NTCP pointed to the possibility that NTCP was the fish we were looking for.

## 7. Fish Verification

We next sought to verify that NTCP was the “big goldfish” from the following aspects: specific interaction, loss of function, the gain of function, and species specificity. To demonstrate the specific interaction between preS1 and NTCP, Huan Yan showed that: similar cross-linking signals could be reproduced based on 293T cells’ exogenous expressing of human NTCP (hNTCP) or treeshrew NTCP; hNTCP expression in Huh-7 cells markedly increased HDV’s binding efficiency; specific interaction and subcellular colocalization between the preS1 peptide and NTCP could be demonstrated by flow cytometry and immunofluorescence assays. As for the loss of function, Guocai Zhong showed that the susceptibility to HBV and HDV could be suppressed by NTCP silencing in PTHs, primary human hepatocytes (PHHs), and HepaRG cells. Gain of function, the gold standard for validating the functionality of a receptor, was first achieved in the HDV infection of huh-7 cells transfected by NTCP by Huan Yan (Figure 3) and later demonstrated by HBV infection in HepG2 cells stably expressing hNTCP (HepG2-NTCP) by Huan Yan and Guocai Zhong. Optimizing the cell culture condition and using a primary hepatocyte maintenance medium containing DMSO was essential to show the receptor functionality of NTCP, particularly for HBV infection. Notably, the gain of susceptibility for hepatoma cells by ectopically expressing the NTCP provided efficient and convenient HBV/HDV infection models that were greatly needed in the field. Lastly, Yan et al. conducted comparative analyses and identified a critical host range determinant restricting monkey NTCP (aa157-165) from supporting HBV/HDV binding and entry. Collectively, these data provided very strong evidence that the NTCP was the big fish that other researchers have been struggling to capture for quite a long time.

## 8. Discussion

Understanding the entry process of HBV and HDV is of great scientific and clinical importance. Over the past few decades, there has been no clear picture of how HBV binds to and enters the hepatocyte. Although a list of studies focused on HBV receptor hunting and resulted in the identification of at least ten HBV preS1 interacting proteins, none of them were functional in rendering viral infection susceptibility. The lack of suitable infection models for HBV and HDV greatly impeded both basic and clinical research on these viruses. Our identification of the HBV receptor NTCP uncovered the mystery and opened new avenues for studying the critical role of NTCP in HBV infection [26]. Later, a study from Stephan Urban’s group separately proved that NTCP is a functional HBV receptor by comparing the gene expression pattern between the differentiated HepaRG cells and the naïve cells, followed by examining HBV infection efficiency after the small hairpin RNA-mediated silencing of NTCP in HepaRG cells and overexpression of NTCP in hepatoma cells [27]. This net-fishing approach does not require a complex bait design and extensive IP experiments, whereas more efforts are necessary to verify multiple candidates without known clear features of the true target. In addition, the HBV host range determination, the residues involved in the interaction between preS1 and NTCP, the crosstalk between HBV infection and bile acid uptake, NTCP-targeting entry inhibitors, and the importance of EGFR in HBV entry have been elucidated in detail over the years [27,28,29,30,31,32,33,34,35,36]. Ten years after our initial reports, four recent studies separately resolved the cryo-EM structure of human NTCP under different states and conditions [37,38,39,40,41], unlocking the mystery of how HBV preS1 peptides bind to NTCP, although the resolution and molecular details remain to be improved. NTCP is now widely accepted to be the bona fide receptor of HBV and HDV. The substantial progress since the identification of NTCP as a functional receptor has brought new hopes for the effective management of Hepatitis B and D with the development of viral entry inhibitors and other anti-viral therapeutics.

## 9. Concluding Remarks

The HBV receptor discovery project lasted for about five years, was initiated by establishing the PTHs system and validating the preS1 functionality, facilitated by the specially designed bait for photo-cross-linking, and achieved through step-by-step progress along with numerous failures. Our team made it to the finish line by finding the right direction (fishing pond, bait design, and other fishing tools) and removing all obstacles on the bumpy road (fish skills and tricks, fish identification and verification). A retrospect of this exciting journey has underscored the importance of critical thinking, creative design, and perseverance in great scientific discoveries. We believe that this “fishing” strategy can also provide useful insights into other research aimed at identifying a protein target with defined bait information.

## Figures and Tables

**Figure 1 viruses-15-00512-f001:**
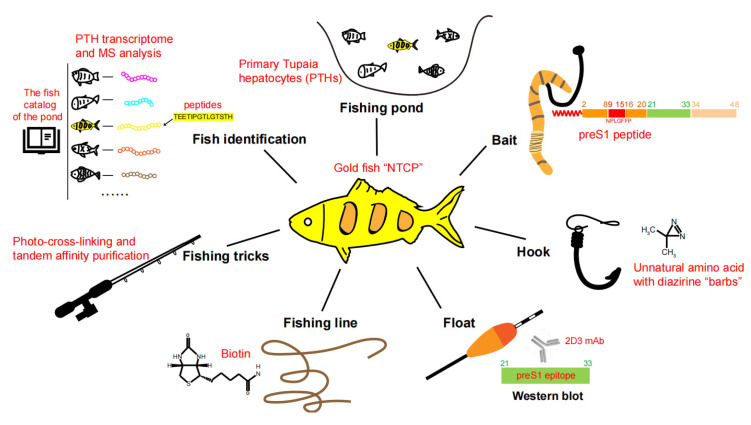
Key factors for identifying NTCP as a functional HBV receptor. The HBV receptor NTCP is compared to a big “goldfish” to be captured. The technical details of the “fishing pond, bait, hook, float, fishing line, fishing tricks, and ways of fish identification”, crucial for NTCP identification, are described in the illustration.

**Figure 2 viruses-15-00512-f002:**
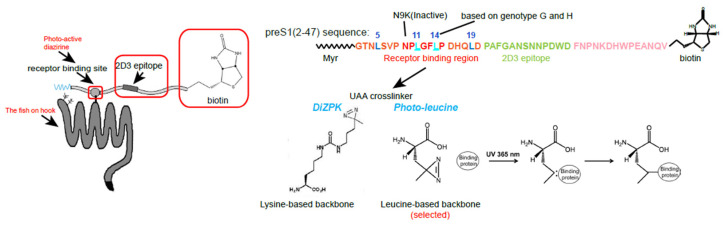
Photo-reactive PreS1 bait peptide design for NTCP identification. The left cartoon shows the strategy of the photo-reactive preS1 bait peptide design and the proposed bait–receptor complex after cross-linking. Arrows indicate receptor binding site, 2D3 epitope and the target. The two affinity tags are highlighted by red boxes. The right panel displays the sequence and modification details of the preS1 bait peptide as well as the N9K control peptide, and the reaction mechanism of the diazirine groups for photo-crosslinking upon 365 nm UV light irradiation. The photo-leucine but not DiZPK were selected for UAA insertion at sites 11 and 14.

**Figure 3 viruses-15-00512-f003:**
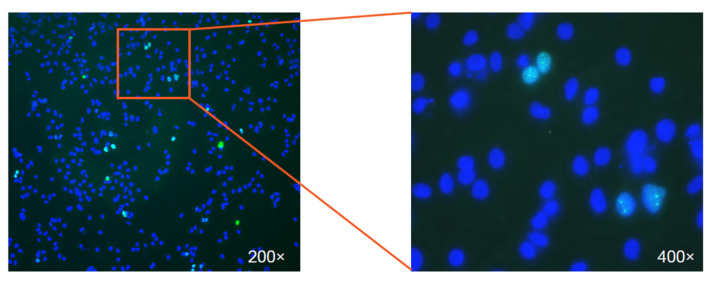
The first view of HDV infection in Huh7-NTCP cells. The image shows the immunostaining of the nuclear-localized HDV delta antigen (green) in HDV infected Huh7 cells expressing human NTCP. The nuclei were stained in blue. The right panel displays an enlarged view of the infected cells.

## Data Availability

No new data were created or analyzed in this study. Data sharing is not applicable to this article.

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
