# Peer review of "Key Factors for “Fishing” NTCP as a Functional Receptor for HBV and HDV"

_viruses, 2023, doi:10.3390/v15020512_

Round 1

Reviewer 1 Report

The review by Yan et al. describes the process of finding an essential entry factor or receptor for the entry of very important human pathogens, namely human hepatitis B and D viruses (HBV/HDV). The text describes several interesting aspects of the process like the choice of cells, crosslinking reagents, purification tags, and more.  

The review is timely, clearly written, and will be of interest to virologists and membrane protein biologists. It also makes an elegant and smart analogy between the receptor discovery process and fishing, which makes the reading fun and pleasant.   

The following are major points that I think should be considered before publication of the manuscript:

-The self-reference of the Elife 2012 article presenting NTCP as the cellular receptor of HBV/HDV is missing in the manuscript (line 60).

-The group of Stephan Urban has published an orthogonal and independent approach and reached the same conclusion about NTCP as the receptor of HBV/HDV (Ni et al., 2014). Although the reference of that article is included in the current manuscript, there is no discussion about potential pros and cons of such alternative approach. In my view, the fact that a competitor group published convergent results a year or so later, only strengthens the results from the Elife 2012 paper, and it should be cited and discussed as such.

-Line 110, it is mentioned that previous studies tried to use PreS1 as a bait, and failed, but there are no explicit references to the articles. Please, include relevant references there.

Minor comments:

-Line 55, “PreS1 binding proteins of known identity…..” shouldn’t it read “unknown”?

-Figure legends should be extended to explain better what it is depicted, particularly regarding Fig. 2 and 3.

Author Response

Yan et al, viruses-2212539

Key factors for "fishing" NTCP as a functional receptor for HBV and HDV

                 -Point-by-point response to the reviews-                   

Reviewer 1

The review by Yan et al. describes the process of finding an essential entry factor or receptor for the entry of very important human pathogens, namely human hepatitis B and D viruses (HBV/HDV). The text describes several interesting aspects of the process like the choice of cells, crosslinking reagents, purification tags, and more.  

The review is timely, clearly written, and will be of interest to virologists and membrane protein biologists. It also makes an elegant and smart analogy between the receptor discovery process and fishing, which makes the reading fun and pleasant.   

Response: We appreciate your positive comments and are happy to know that you found this review interesting.

The following are major points that I think should be considered before publication of the manuscript:

-The self-reference of the Elife 2012 article presenting NTCP as the cellular receptor of HBV/HDV is missing in the manuscript (line 60).

Response: Thanks for the reminder. We have included the missing reference in the revision (line 65).

-The group of Stephan Urban has published an orthogonal and independent approach and reached the same conclusion about NTCP as the receptor of HBV/HDV (Ni et al., 2014). Although the reference of that article is included in the current manuscript, there is no discussion about potential pros and cons of such alternative approach. In my view, the fact that a competitor group published convergent results a year or so later, only strengthens the results from the Elife 2012 paper, and it should be cited and discussed as such.

Response: Many thanks for the comments. We totally agree with you. A short discussion of this important study has been included in the Discussion section.

Revised manuscript-Line 283-290:

Later, a study from Stephan Urban's group separately proved that NTCP is a functional HBV receptor by comparing the gene expression pattern between the differentiated HepaRG cells and the naïve cells, followed by examining HBV infection efficiency after small hairpin RNA-mediated silencing of NTCP in HepaRG cells and overexpression of NTCP in hepatoma cells [27]. This net-fishing approach does not require complex bait design and extensive IP experiments, whereas more efforts are necessary for verifying the multiple candidates without known clear features of the true target.

-Line 110, it is mentioned that previous studies tried to use PreS1 as a bait, and failed, but there are no explicit references to the articles. Please, include relevant references there.

Response: Thanks for reminding us. We have cited relevant references in the revision (Line124).

Minor comments:

-Line 55, “PreS1 binding proteins of known identity…..” shouldn’t it read “unknown”?

Response: Thanks for pointing it out. We've corrected it to "unknown".

-Figure legends should be extended to explain better what it is depicted, particularly regarding Fig. 2 and 3.

Response: Thanks for the suggestion. Legends with more details have been included in the revision to assist better understanding.

Reviewer 2 Report

This review is an artistic description of the key discovery in the field of HBV and HDV of the past decade by one of its major players, and the first author of the prominent eLife publication. The authors provide a narrative of their 5 years long journey in the identification and verification of NTCP as the major, functional receptor, required for HBV/HDV binding and entry. The story tells about the process of probing, frustration, and, finally, the success in the identification of NTCP by mass spectrometry analysis. Overall, this is an interesting story that is useful both as a guide for people investigating viral receptors, and for understanding the key steps in the major discovery of NTCP in HBV/HDV life cycle.

I have only several issues that are unclear to me:

(1)    The key paper is not cited here for some reason, that is the eLife publication where NTCP was first described as a receptor for HBV;

(2)    Both authors tell the story of NTCP discovery, but the second author of the paper is not in the list of eLife co-authors. Thus, probably the presentation of the material from the position of discoverers should be re-phrased?

Other minor issues are outlined below:

Line 111-112: references missing

Line 139: is it appropriate to state “receptor binding activity” here, while you had not yet identified the receptor?

Line 177: the authors use “I” noun, but two authors are writing this review. Please, either specify or re-phrase

Line 182: spaces are missing in 500nM and 250nM

Author Response

Yan et al, viruses-2212539

Key factors for "fishing" NTCP as a functional receptor for HBV and HDV

                 -Point-by-point response to the reviews-                   

Reviewer 2

This review is an artistic description of the key discovery in the field of HBV and HDV of the past decade by one of its major players, and the first author of the prominent eLife publication. The authors provide a narrative of their 5 years long journey in the identification and verification of NTCP as the major, functional receptor, required for HBV/HDV binding and entry. The story tells about the process of probing, frustration, and, finally, the success in the identification of NTCP by mass spectrometry analysis. Overall, this is an interesting story that is useful both as a guide for people investigating viral receptors, and for understanding the key steps in the major discovery of NTCP in HBV/HDV life cycle.

 Response: We are grateful to this reviewer for considering this review interesting and for his/her acknowledgment of the significance of this discovery.

I have only several issues that are unclear to me:

  • The key paper is not cited here for some reason, that is the eLife publication where NTCP was first described as a receptor for HBV;

Response: Thanks for the reminder. We did cite the eLife paper, but not in the right place. We placed the missing reference when it first appeared in the revised manuscript (Line 65).

  • Both authors tell the story of NTCP discovery, but the second author of the paper is not in the list of eLife co-authors. Thus, probably the presentation of the material from the position of discoverers should be re-phrased?

Response: Thank you for pointing it out. We have re-phrased related content to specify the contribution of the discoverers to avoid potential misunderstanding. For example, we changed “we” in the first sentence of the abstract to “Wenhui Li's research group”. In addition, the statement “The work of identifying NTCP as the functional HBV receptor was led by Wenhui Li, and was accomplished by a team spearheaded by Huan Yan and with critical contributions from Guocai Zhong (co-first author) and many other colleagues at NIBS, Beijing. ” was moved from the “Concluding remarks” to the “Introduction” section to clarify who made the discovery.

Other minor issues are outlined below:

-Line 111-112: references missing

Response: Thanks for reminding us. We have included relevant references here in the revision (Line124).

-Line 139: is it appropriate to state “receptor binding activity” here, while you had not yet identified the receptor?

Response: Many thanks to the reviewer for his/her advice. We have re-phased it to "PTH binding activity" (Line153).

-Line 177: the authors use “I” noun, but two authors are writing this review. Please, either specify or re-phrase

Response: Thanks for pointing this out. We have re-phrased the related content.

Revised manuscript-Line 193-196:

Before giving up, Huan Yan wondered if he had put too much bait into the cells so that the non-specific cross-linking signal covered the specific signal. He then conducted a PTHs cross-linking experiment using serial-diluted bait peptides.

-Line 182: spaces are missing in 500nM and 250nM

Response: Thanks to the reviewers for their careful review. We have corrected it. (Line198-199)
